# Experimental Study on Gaseous Moisture Absorption and Swelling of Red-Bed Mudstone in Central Sichuan, China under Different Relative Humidity Environments

Fei Yu [1], Kaiwen Tong [1,2,*], Zhenghao Fu [3], Zhangjun Dai [1], Jian Li [1] and Kang Huang [1,2]

[1] State Key Laboratory of Geomechanics and Geotechnical Engineering, Institute of Rock and Soil Mechanics, Chinese Academy of Sciences, Wuhan 430071, China; fyu@whrsm.ac.cn (F.Y.)
[2] University of Chinese Academy of Sciences, Beijing 100049, China
[3] School of Urban Construction, Wuhan University of Science and Technology, Wuhan 430065, China
* Correspondence: tongkaiwen19@mails.ucas.edu.cn; Tel.: +86-18801068440

**Abstract:** Aiming to resolve the engineering hazards caused by the long-term micro-swelling of red-bed mudstone, gaseous moisture absorption-swelling tests under four ambient humidities were conducted (60%, 70%, 85% and 99%) by developing a set of experimental devices. Based on the moisture absorption characteristics of mudstone, the effects of a vapor pressure gradient on the equilibrium time, moisture absorption rate and swelling rate were discussed. Combined with its mineral composition and pore structure, the correlation mechanism between gaseous moisture absorption and swelling was explained in depth. The experimental results suggested that the whole process was long-term and slow. The swelling rate was lower than the moisture absorption rate by two orders of magnitude. As a result, the duration of a stable swelling strain was two orders of magnitude higher than that of a stable moisture absorption. The characteristic curve of moisture absorption went through three stages: the rapid stage, dominated by crystal layer adsorption; the intermittent expansion stage, dominated by adsorbed water film; and the decelerated stage, dominated by capillary condensation. In the intermittent expansion stage, as the relative humidity declined, the duration of an intermittency increased, and the number of intermittencies decreased. The evolution of the swelling rate obviously lagged the proportion of moisture absorption. This hysteresis effect reached its maximum when the moisture absorption transitioned from the intermittent expansion stage to the decelerated stage. Moreover, the equilibrium time, moisture absorption ratio and swelling strain all had a non-linear proportional relationship with the relative humidity. In this case, the critical value was 85%. Finally, a unified mathematical expression for the gaseous moisture absorption curve was acquired and the relationship between the swelling strain and the increment of moisture content was fitted; this will provide the necessary research basis for a subsequent simulation of its volume change under changing ambient humidity.

**Keywords:** red-bed mudstone; gaseous moisture absorption; swelling; relative humidity

## 1. Introduction

Red beds generally refer to a set of strata mainly composed of clastic rocks and argillaceous rocks formed in the Mesozoic and Tertiary continental lakes, rivers and foothills. The Sichuan Basin is one of the most concentrated distribution areas of red beds. For the Jurassic-Cretaceous lacustrine mudstone in this area, the content of hydrophilic minerals such as montmorillonite and illite is relatively high, resulting in special inherent properties such as disintegration, swelling and modification [1,2]. Therefore, it has become a typical representative of disasters and a key research object.

Red-bed mudstone has poor water stability. Its swelling deformation caused by changes in the water environment is the main cause of engineering damage. In this regard, researchers have carried out a lot of research work. Zhang et al. [3] found that the water

absorption of mudstone in the laboratory was significantly higher than the saturated water content in the natural environment. Fan et al. [4] studied the change in saturation of expansive mudstone during swelling and shrinking. Ma et al. [5] and Wang et al. [6] explored the relationship between the permeability properties and deformation through in situ swelling tests. Diao et al. [7], Tang et al. [8] and Yin et al. [9] discussed the effect of temperature on water absorption-swelling and the shear responses of samples. Zhang et al. [10] analyzed the evolution of strength during swelling. Guo et al. [11] focused their research on the strength softening characteristics after a water–rock interaction. With continuous deepening, the research focus has changed from the macro level to the micro level. Yang [12] used CT technology to describe the changes of meso-structure with water immersion time. Al-Rawas et al. [13] and Liu et al. [14] explained the intensity attenuation characteristics by analyzing the SEM images of samples under different water contents.

Most of the above-mentioned research topics are oriented toward the project construction stage. Under the drastic change in the water and stress environment in which the mudstone exists, the swelling and disaster-causing process it undergoes is generally relatively rapid. However, there are few studies on the long-term micro-expansion of mudstone in the case of small environmental gradient changes during the project operation stage [15]. Its non-convergent swelling has caused great safety hazards to project operations, which has led to speed limits for many high-speed rails and huge renovation costs [16–18].

To this end, some researchers conducted a preliminary discussion. Zhang et al. [19] carried out a large-scale swelling test on the high-speed railway mudstone foundation. Wang et al. [20] expounded the deformation law of the foundation under the condition of unidirectional and multidirectional flooding. Dai et al. [21] explored the cause of the long-term arching of the foundation based on field monitoring data. Chen et al. [22] observed the fracture development characteristics of mudstone during water immersion. Zhong et al. [23] studied its deformation time-dependent characteristics through laboratory experiments. Jing et al. [24] revealed the influence mechanism of the hydraulic properties of red beds on the deformation of ballastless track. Although the above studies have well reflected the swelling of red-bed mudstone over time, the internal mechanism of its long-term (several years) slow arching has not been explained clearly. The main reason for this is that the previous works mainly focused on its deformation behavior under direct contact with liquid water, such as complete immersion or partial immersion swelling, while the research on the continuous deformation affected by gaseous water has lacked attention. Additionally, the above work focuses on the time-dependent deformation of mudstone, but the interaction between clay minerals and water is fundamental. Unfortunately, the variation in water content during the water absorption process of mudstone has not received sufficient attention.

In actual engineering, the permeability coefficient of mudstone is very small. The liquid water that exists only in primary fractures and interlayer interfaces has poor connectivity, thereby the water-wetting range is limited [25–27]. In addition, the mudstone is mostly in a dry unsaturated state and the flow of water vapor becomes the main form of pore water migration. In fact, vapor flow occurs in any direction at the atmosphere–sample interface, specifically from one point to another in the pore space. On the one hand, the small porosity of mudstone determines that the diffusion of steam in the pores is particularly difficult, so the interaction between water vapor and clay minerals may be a continuous and slow process. On the other hand, the vapor pressure in the pores tends to fluctuate repeatedly with changes in ambient air pressure and temperature, which also has an important impact on long-term micro-swelling deformation to a certain extent. Regarding the gaseous moisture absorption of rocks, researchers have also completed some initial experiments [28,29], but mainly on the effect of the amount and rate of moisture absorption. Furthermore, for the experimental setup of gaseous moisture absorption, researchers mainly maintain a tight contact between the iron hoop or lateral confinement and the specimen under atmospheric conditions to induce a vertical deformation. However, it cannot control the relative humidity and lateral water absorption, leading to a possi-

bly uneven water distribution. In effect, the research on the swelling characteristics of red-bed mudstone during gaseous moisture absorption is basically non-existent. Further, it is difficult to understand effects of the mechanism of gaseous moisture absorption on long-term deformation.

In the current work, the red-bed mudstone in central Sichuan, China was taken as the research object. By carrying out gaseous moisture absorption-swelling experiments under different relative humidity environments, the process of gaseous moisture absorption and swelling deformation, as well as the coordination of the two, were first analyzed. Secondly, the migration characteristics and volume changes of the water vapor phase in the pores were discussed. Further, the influence of the gradient of vapor pressure (relative humidity) on the whole process was studied. Finally, combined with the mineral composition and pore structure, the time-dependent characteristics and the long-term micro-expansion mechanism were deeply elucidated.

## 2. Physical and Mechanical Properties of the Red-Bed Mudstone

### 2.1. Stratum Lithologic

The samples were taken from the Weiyuan section of the ChengZiyi high-speed railway, which is located in the gentle and low fold belt of central Sichuan. It has a monoclinic structure with nearly horizontal rock formations. The overlying slope of the residual $(Q_4^{dl+el})$ expansive soil is about 0~2 m thick. The underlying bedrock is the Shangshaximiao formation of the Jurassic middle series ($J_2s$) of mudstone interbedded with sandstone. As shown in Figure 1, the mudstone that was freshly excavated on site has a complete appearance and no joints or fissures. It has medium-thick layers. Moreover, the sample is brownish red with a slight bluish gray. Oily substances are distributed on the surface, which is muddy cementation. In the natural state, it tends to disintegrate, soften, shrink and crack when encountering water.

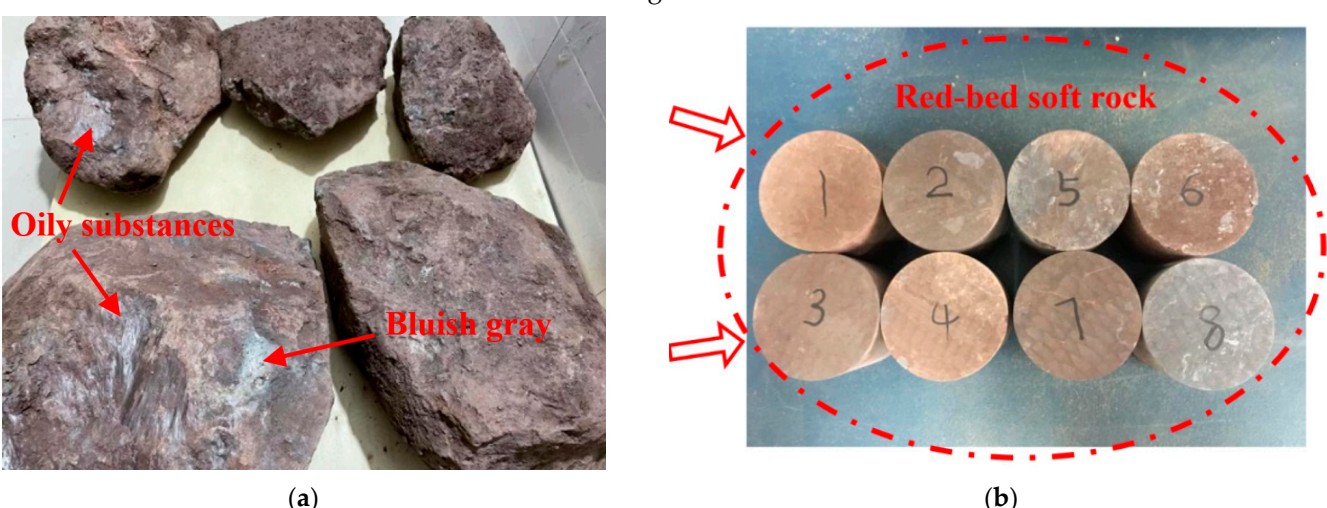

|     |     |
| :-: | :-: |
| (**a**) | (**b**) |

**Figure 1.** Red-bed mudstone before and after processing: (**a**) excavated irregular red-bed mudstone; (**b**) processed standard cylindrical samples (diameter: 50 mm, height: 50 mm).

### 2.2. Mineral Composition and Microstructure

As a typical expansive rock, the expansibility of mudstone depends on the swelling between the clay particles or crystal layers. The higher the clay content, the greater the swelling strain. To obtain the mineral composition of red-bed mudstone, especially for expansive clay minerals such as montmorillonite, illite and kaolinite, 10~20 g samples were selected for X-ray diffraction (XRD) after drying, grinding and passing through a 0.05 mm sieve. Table 1 lists the XRD results. Among them, the clay minerals are mainly montmorillonite and illite, accounting for 19.67–20.14% and 14.54–15.36%, respectively. This provides the necessary material basis for the swelling and deformation of the sample during water absorption.

**Table 1.** XRD test results of red-bed mudstone.

| Sample Group | Mineral Content (%) | | | | | | |
|---|---|---|---|---|---|---|---|
| | Montmorillonite | Illite | Quartz | Albite | Kaolinite | Calcite | Hematite |
| 1 | 19.67 | 15.36 | 35.32 | 18.11 | 5.12 | 4.75 | 1.66 |
| 2 | 20.14 | 14.54 | 38.63 | 20.54 | 4.88 | — | 1.27 |

Figure 2 corresponds to the SEM results of the samples. On the microscopic scale, the clay minerals of red-bed mudstone are mainly distributed in the form of flakes and irregular sheet-like stacks. The internal pores of the sample are well developed. There are a certain number of microcracks and fissures.

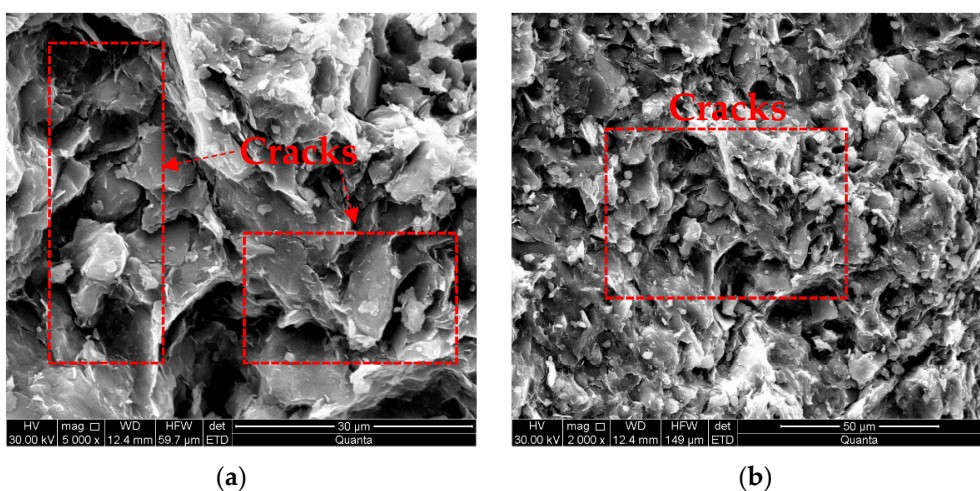

(**a**)  (**b**)

**Figure 2.** Microstructure of mudstone: (**a**) 5000 times; (**b**) 2000 times.

*2.3. Structural Characteristics of Pores and Fissures in Initial Sample*

To further explore the scale range, type and quantitative parameters of the pore-fracture structure, the results of X-ray tomography (μ-CT), mercury injection technique (MIP) and low temperature nitrogen adsorption (BET) were utilized. Referring to the previous test results [30], the pores and fractures of mudstone were tested in sequence according to the scale range: CT scanning was used to analyze pores larger than 100 μm. Mercury intrusion testing was carried out for pores of 100 nm~100 μm, while the parameters of pore diameters less than 100 nm were derived from low-temperature nitrogen adsorption tests (Figure 3).

The results showed that the primary fissures identified by CT scanning were barely developed. Fracture volume accounted for only 0.17%. The connectivity rate was 0.09% and the diameter-to-length ratio of the pores was 0.01~0.58. The above data suggests that the structure of the undisturbed mudstone was compact, with a few primary fractures and mostly closed and semi-closed flat fractures. At the nano and micro levels, combined with a BET test and MIP data, the pore volume of undisturbed mudstone was about 8.99 mm$^3$/g, and the calculated porosity was 2.32%. Specifically, 90% of the pores had a pore size smaller than 2 μm. The peak of the pore size was located at 40 nm, corresponding to the interlayer spacing of layered clay minerals such as montmorillonite and illite. For micropores smaller than 10 nm, the pore volume accumulated fastest in the range of 1~2 nm, which may be equal to the interplanar spacing of clay.

To sum up, there are few primary cracks in the sample on the macro scale. Likewise, there are fewer intergranular pores (>10 μm) on the microscopic scale. The interlayer pores of clay minerals are the main pore type on the nanoscopic scale and are enriched in two pore size ranges of 1~2 nm and 40 nm, corresponding to the parallel crystal plane spacing and non-parallel layer spacing of clay minerals, respectively. Therefore, these samples with undeveloped primary fissures can be used to carry out subsequent experiments.

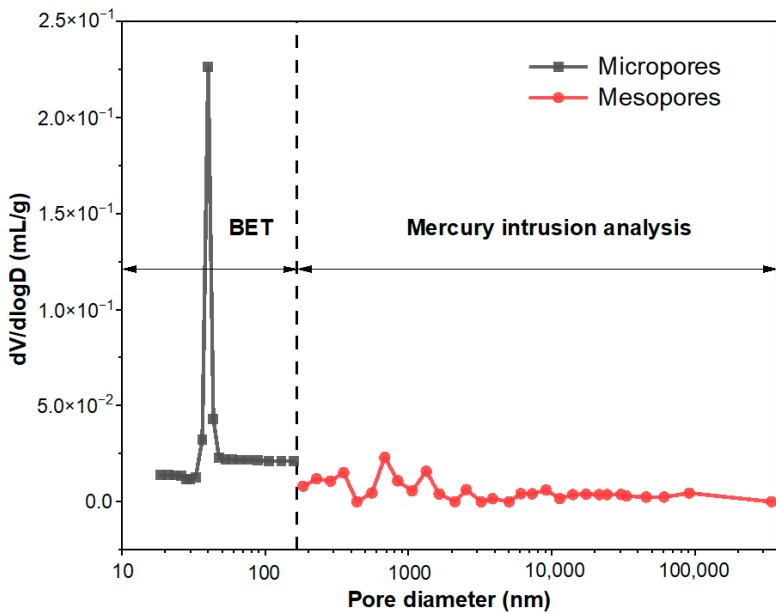

**Figure 3.** The distribution curve of joint pores of red-bed mudstone [30].

*2.4. Evaluation of Swelling Properties of Undistributed Rock*

To clarify the natural state and swelling performance of red-bed mudstone, a series of physical and mechanical tests were carried out, including free swelling strain test of rock powder, confined vertical swelling test and vertical swelling force test. The swelling force was tested using the constrained deformation method instead of the traditional back pressure method which has a large measurement error for structural mudstone with high stiffness. The swelling strain and swelling force tests were all completed under the condition of complete immersion in water.

The whole experimental process was strictly in accordance with "Draft Recommended ISRM Methodology for Laboratory Testing of Expansive Rocks" proposed by Madsen et al. [31], "Specification of soil test" SL237-1999, "Code for Rock Test of Railway Engineering" TB10115-2014 and "Specification for rock tests in water conservation and hydroelectric engineering" SL264-2016. The test results are listed in Table 2.

**Table 2.** Basic physical parameters of red-bed mudstone.

| Project (Weiyuan) | Natural Density (g/cm$^3$) | Natural Water Content (%) | Saturated Absorbed Water Ratio (%) | Free Swelling Strain (%) | Vertica Swelling Strain (%) | Vertical Swelling Force (kPa) |
|---|---|---|---|---|---|---|
| Number of samples | 14 | 14 | 11 | 11 | 4 | 7 |
| Maximum | 2.62 | 3.40 | 11.48 | 29.2 | 3.42 | 1558.9 |
| Minimum | 2.54 | 2.60 | 6.07 | 6.3 | 0.78 | 152.3 |
| Average | 2.58 | 2.77 | 9.47 | 13.2 | 1.99 | 739.1 |

According to Table 2, the equilibrium water content of mudstone in the natural environment is low, with an average value of only 2.77%, which is much lower than the average level of 9.47%. This means that the original rock sample has a large water-rich space. The average free swelling strain of the rock powder sample and the average vertical swelling strain are 13.2% and 1.99%, respectively, indicating that the deformation of the original rock is relatively small.

Furthermore, the original rock density is high. The average value reached 2.58 g/cm$^3$ and the porosity was only 2.86%. Its deformation modulus is significantly higher than that of most expansive soils. In this case, a small swelling deformation will generate a large swelling force when it is confined. This has been fully verified in the vertical swelling

force test. The average swelling force was 739 kPa and the maximum value even exceeded 1.5 MPa, which is enough to cause damage to engineering structures.

## 3. Gaseous Moisture Absorption-Swelling Test

In the gaseous moisture absorption-swelling test, it is necessary to equilibrate the sample in a stable ambient humidity for a long time. For this purpose, the saturated electrolyte salt solution will be used to create a test space with a controlled relative humidity. The matrix suction at a certain temperature can be determined by the relative humidity, thus allowing suction control by adjusting the ambient humidity.

According to the instructions of the international organization OIML [32,33] for saturated solutions and standard relative humidity, the ratio of the volume of the constant humidity container to the surface area of its walls should be as small as possible to better maintain a stable humidity in the enclosed space. Based on this principle, the humidity control box was designed. The size of the box was 500 mm × 500 mm × 700 mm. It is made of highly transparent polypropylene material, as shown in Figure 4. The saturated salt solution was placed along the bottom of the box with a sample holder in the center to ensure that the sample was under a stable vapor pressure. The test box adopted a double-layer sealing method. The lower and the upper sealing layer are, respectively, a transparent plastic wrap and a high transparent top cover. The purpose of this was to minimize the humidity exchange between the air in the box and the outside air. The temperature of the experiment was kept at 25 °C during the experiment. Through the highly sensitive temperature-humidity meter, the ambient temperature and humidity can be observed in real time. On the sample holder of swelling strain, the dial gauge can monitor small volume changes and the fast camera equipment is able to feedback the deformation of the sample quickly. There is a high-precision weighing device on the top cover of the box to record the change in quality of the rock sample during the test.

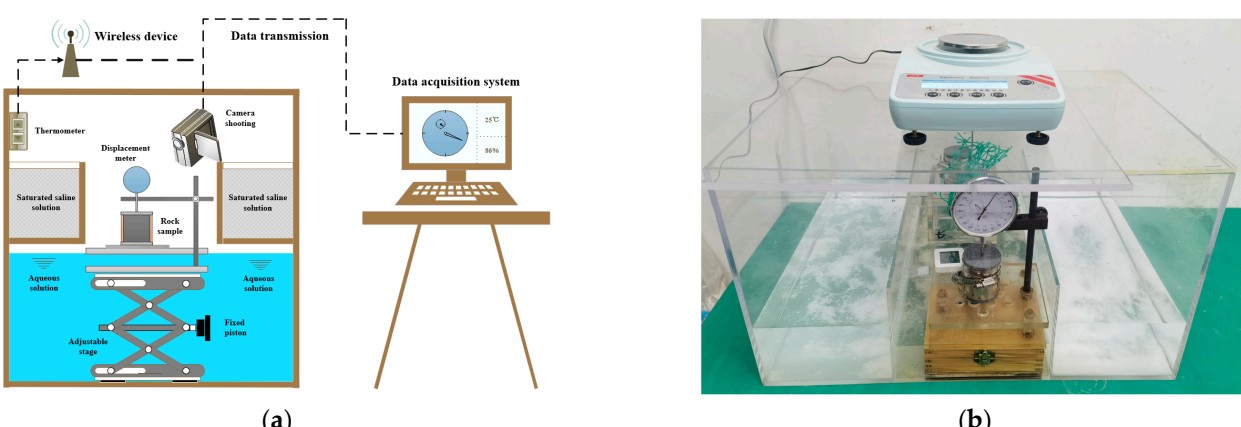

**(a)**                                              **(b)**

**Figure 4.** Test device of gaseous moisture absorption-swelling: (**a**) schematic device diagram; (**b**) real device.

Rock samples were prepared from the same batch of intact rock blocks by wire cutting. The diameter and height of the sample were both 50 mm (Figure 1). To obtain the full moisture absorption process and avoid the influence of the difference in initial water content, the samples were dried in an oven at 65 °C to constant weight. They were then cooled naturally in a desiccator for 24 h. The original sample after drying had no obvious cracks and was complete, which may be related to its low water content in the natural state. The dried samples were confined by a steel hoop to limit their lateral deformation. At the same time, a certain number of round holes with a diameter of 5 mm and a spacing of 3 mm were put onto the steel hoop to facilitate side gaseous moisture absorption. The device mainly monitored the axial swelling strain.

Referring to the *RH* data of the saturated salt solution given by the International Legal Organization [32], saturated $K_2SO_4$ solution, KCl solution, KI solution and NaBr solution

were selected to generate four different *RHs*, corresponding to 99%, 85%, 70% and 60%, respectively (Table 3). The matric suction can be calculated using the Kelvin equation:

$$u_a - u_w = -\frac{RT\rho_w}{M_w} \ln RH \tag{1}$$

where $u_a$ is the total air pressure; $u_w$ here refers to the pore water pressure; and $u_a - u_w$ is the matrix suction. The ideal gas constant $R$ is 8.314 J·mol$^{-1}$·K$^{-1}$. T at 298.15 K. The density, $\rho_w$, of water is 1000 kg/m$^3$. The molecular weight $M_w$ of water is 0.018 kg.mol$^{-1}$. Accordingly, the matric suction gradually increased from 1.38 MPa in the *RH* = 99% environment to 70.34 MPa in the *RH* = 60% environment.

**Table 3.** Environmental parameters and sample parameters under different *RH*.

| RH/% | Saturated Salt Solution | Environmental Temperature/°C | Matric Suction /MPa | Initial State of Sample | Sample Number |
|---|---|---|---|---|---|
| 99 | K$_2$SO$_4$ | | 1.38 | | XS-1, PZ-1 |
| 85 | KCl | | 22.32 | 65 °C | XS-2, PZ-2 |
| 70 | KI | 25 | 48.98 | drying sample | XS-3, PZ-3 |
| 60 | NaBr | | 70.34 | | XS-4, PZ-4 |

## 4. Results and Discussion

### 4.1. Gaseous Moisture Absorption-Swelling Law

Considering that the moisture absorption-swelling process of mudstone under different *RHs* was generally similar, the test results under 99% *RH* will be analyzed in detail.

#### 4.1.1. Changes in Moisture Absorption Rate over Time

Considering that the moisture absorption-swelling process of mudstone under different RHs was generally similar, the test results under a 99% *RH* will be analyzed in detail.

The moisture absorption characteristic curve of dried mudstone in an environment with a 99% *RH* is presented in Figure 5. For a normalized comparison, the ordinate (moisture absorption ratio) in Figure 5 is the ratio of its changes in total mass ($Q = m - m_0$) to its initial mass $m_0$. In terms of the initial state of the drying sample, it is actually equal in moisture content, w, to the dried sample.

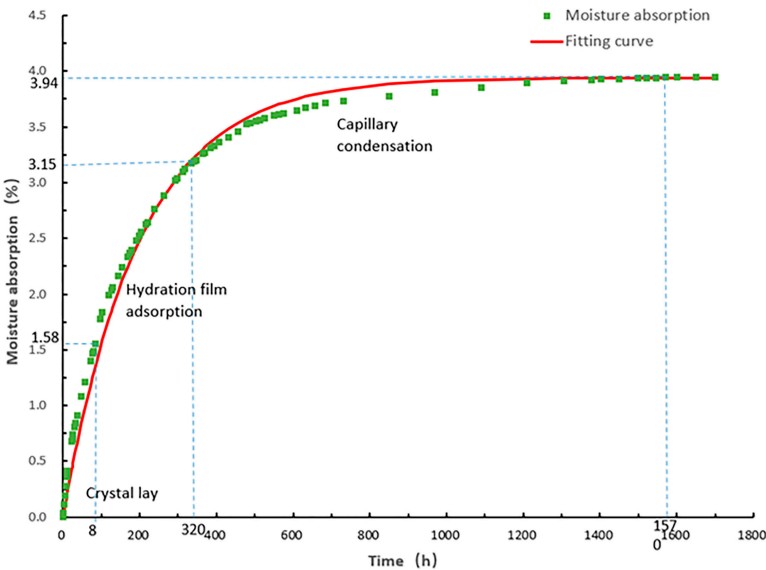

**Figure 5.** The characteristic curve of moisture absorption ratio changing with time under 99% *RH*.

In the process of gaseous moisture absorption, the moisture absorption ratio of mudstone increased with time (Figure 5). In general, it presented a decelerating growth trend. The curve was relatively smooth and there was no obvious inflection point, which can be fitted with a first-order exponential decay function:

$$w(t) = 3.94 - 3.94\mathrm{e}^{-0.005t}, R^2 = 0.99 \tag{2}$$

In the formula, $w(t)$ is the gaseous moisture absorption ratio within 0~$t$. According to Figure 5, the fitting curve basically coincides with the experimental data.

As opposed to with complete water immersion and capillary moisture absorption, the change in water absorption rate under gaseous moisture absorption was a long-term and slow process [30]. Under a 99% *RH*, the sample reached stability after 1570 h. The stable moisture content was 3.94%, which is greater than the natural moisture content of 2.77%. It indicates that natural mudstone was in a gaseous moisture absorption equilibrium stage and the equilibrium humidity was lower than 99% *RH*, which has a high potential for water adsorption.

The slope of the curve in Figure 5 was initially large but then gradually tended to be gentler. In Figure 6, its moisture absorption rate d$w$/d$t$ as a function of time is plotted. According to the overall development of the curve, this process can be roughly divided into three stages: rapid moisture absorption, decelerated moisture absorption and slow moisture absorption.

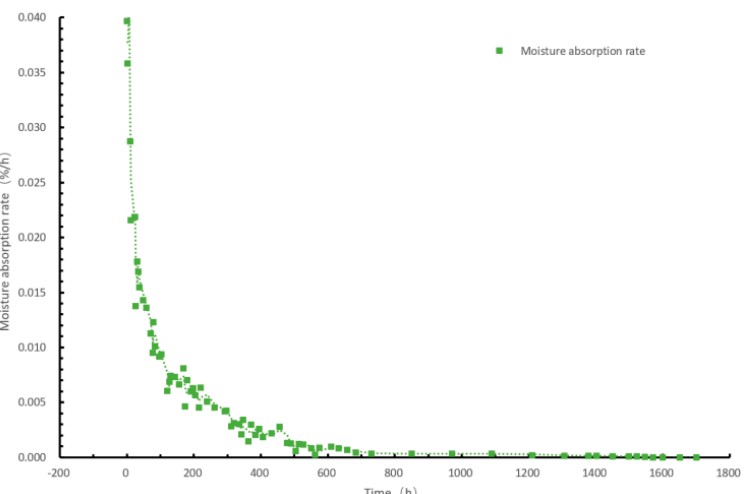

**Figure 6.** Moisture absorption rate over time under 99% *RH*.

In the rapid moisture absorption stage, 40% of the total moisture absorption was completed in 84 h, corresponding to a water content of 1.58% and an average moisture absorption rate of 0.019%/h. At this time, the slope of the curve changed less and was approximately a straight line. The reason for this may be that the rock samples were initially dry and the content of montmorillonite and illite was high. Clay minerals have extremely low free energy in a dry state, much lower than the chemical potential energy of water vapor in the surrounding environment. Referring to the balance principle of chemical potential energy, water vapor with a higher potential energy will migrate to rock samples with a lower potential energy. Therefore, gaseous water molecules will be strongly adsorbed into the mineral crystal layer to balance the potential energy between layers. For example, the interplanar distance $d_{001}$ of dry montmorillonite is mostly less than 1.49 nm, while the suction is as high as 117 MPa or more [34], which is much higher than the suction level of 1.38 MPa at 99% *RH*. Under this high suction gradient, this stage can also be called the "adsorption stage of the crystal layer".

In the decelerated moisture absorption stage, 40% of the total moisture absorption was completed in 84~320 h (accounting for 15.03% of the total time), corresponding to a water

content of 3.15% and an average moisture absorption rate of 0.0067%/h. At this stage, as the potential energy inside the crystal layer is gradually balanced, gaseous water begins to be adsorbed on the particle's surface and exists in the form of a thin water film. This process is controlled by short-range interactions between the water vapor and clay surface, such as polarized electric fields and the Van der Waals attraction. Since the short-range force is smaller than the intermolecular force, the rate of adsorption of water molecules is also slow. Moreover, the short-range force decays rapidly with the increase in the distance between particles, causing the moisture absorption rate to decrease with the thickening of the water film. This stage can be called the "adsorption of water film on particle surface".

In the slow moisture absorption stage, 20% of the total moisture absorption was completed within 320~1570 h (accounting for 79.62% of the total time), with an average moisture absorption rate of 0.0006%/h and a stable water content of 3.94%. When the film absorbs water to a certain thickness, the short-range adsorption is weak. Due to the small distance between particles or the aggregates of mudstone (the pore size is mainly 2~50 μm), higher capillary suction will be promoted in the pores. Capillary condensation will occur if it is higher than the corresponding suction of 99% *RH*. This phase transition and the final stable state can be described by the Young–Laplace equation [35]:

$$P^L = -\gamma_l \left( \frac{1}{R_1} + \frac{1}{R_2} \right) \tag{3}$$

where $P^L$ and $\gamma_l$ are the Laplace pressure and the liquid–air interfacial energy, respectively. $R_1$ and $R_2$ represent the principal radii of curvature of the meniscus, which can be simplified as the pore radius for a sphere-like pore. Further, combined with Equation (1), the Laplace pressure can be given as:

$$P^L = -\frac{R_g T}{V_L} \ln RH \tag{4}$$

where $R_g$, $T$ and $V_L$ are the gas constant, absolute temperature and vapor molar volume, respectively. The adsorbed water at this stage is essentially the capillary condensed water determined by the pore structure. Its adsorption strength is much weaker than the molecular force between crystal layers and the short-range force on the particle surface, resulting in a slow capillary condensed moisture absorption rate. If capillary condensate gradually fills the pores and eventually equilibrates with the surrounding vapor pressure to form a stable gas–water meniscus, moisture absorption will stop. This stage can be called "capillary condensed adsorption in intergranular pores".

In summary, for a complete mudstone sample, the actual moisture absorption process is gradually promoted from the surface to the interior. The moisture absorption process of particles and pores in different parts is not synchronized. In fact, in each hygroscopic stage, the above three adsorptions occur simultaneously and cannot be absolutely divided into several time stages. The reason for this division in this paper is to clarify that the dominant adsorption mechanism is different in each stage, which is also the main internal cause of the change in the moisture absorption rate.

### 4.1.2. Changes in Swelling Strain over Time

The development curve of the swelling strain δ of the dried sample at 99% *RH* with time is shown in Figure 7. During gaseous moisture absorption, the swelling strain increased with time, after which the swelling strain decreased gradually. Overall, it was still a long-term and slow process. It took more than 2 months (1546 h) for the rock sample to expand stably, and the stable expansion rate was 0.406%. Figure 7 shows that the swelling strain was larger and the average $\mathrm{d}\delta/\mathrm{d}t$ was 0.0007%/h in the first 84 h. This stage reached 15.7% of the total swelling strain within 5.43% of the total duration, which can be classified as the "rapid swelling stage". The reason may be that water molecules are adsorbed between clay sheets, resulting in the enlargement of particle volume.

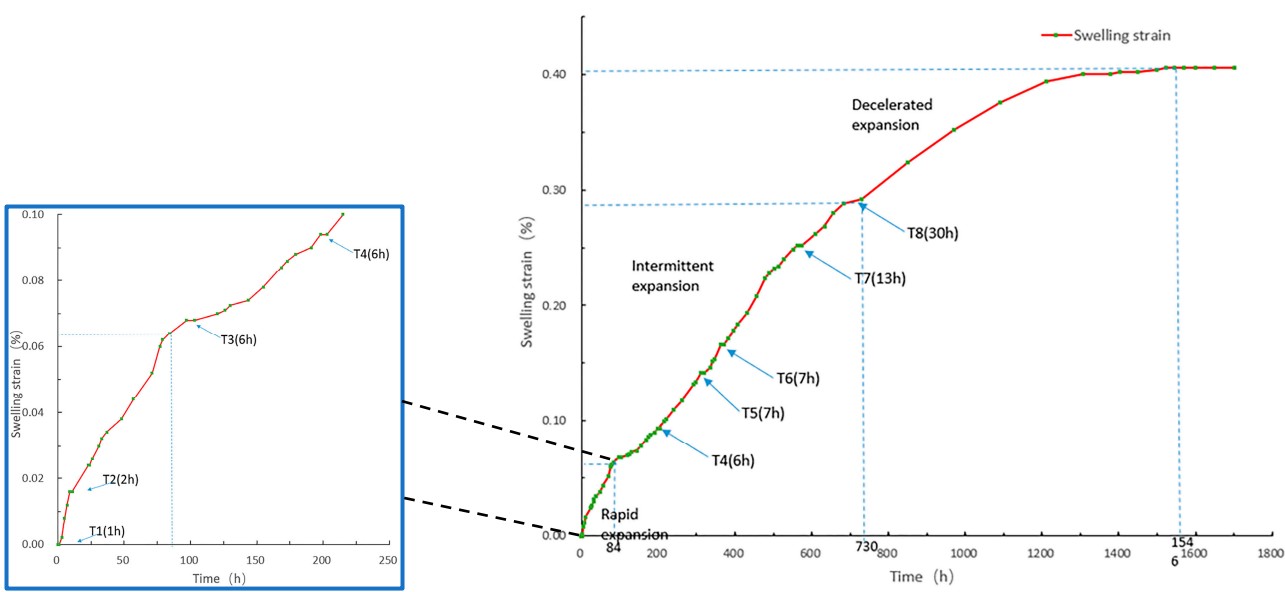

**Figure 7.** The swelling strain of dried sample at 99% *RH* with time.

The curve for 84~730 h was close to a straight line, indicating that the swelling strain changed little. Its average value was about 0.00035%/h, which is half of that in the rapid swelling stage. It is worth noting that the curve at this stage was not smooth but stepwise (T1–T8 in Figure 7). The length of the steps gradually enhanced, which means that the deformation was discontinuous, and the duration of this transient stability gradually became longer. In view of this, it can be called "intermittent expansion". This took 646 h (accounting for 41.78%) corresponding to 56.2% of the total deformation and the swelling strain was 0.29%.

The rapid swelling stage also develops in steps, but with shorter intervals. This phenomenon may be related to the discontinuous variation of interplanar spacing. Because the interplanar spacing ($d_{001}$) of fully dry montmorillonite is 0.96 nm, while the $d_{001}$ under one and two layers of water molecules are 1.25 nm and 15.5 nm, respectively [34,36].

In Table 4, the interval duration and water absorption rate of the T1–T8 period were counted. Combined with Figure 7, it was found that the moisture absorption rate was the largest in the T1 stage at the beginning, but there was no volume change. The swelling occurred after 1 h, indicating that there was an energy accumulation process of water molecules adsorbed into the crystal layer. Subsequently, the duration of each interval was gradually enhanced, which was associated with a decrease in the moisture absorption rate (Figure 8). As mentioned above, the water film on the surface of particles began to appear during the decelerated moisture absorption stage. Intergranular swelling formed as the absorbed water film thickened, which controlled the deformation together with the swelling of the crystal layer. In contrast, intergranular swelling did not have jumps and its deformation was continuous. Since the duration of the continuous deformation increased with fewer intermittent points, it is shown from another perspective that the contribution of the swelling of the crystal layer to the overall deformation declined, while intergranular swelling became the dominant factor.

After the T8 stage, the curve became smooth, and the swelling strain slowed down. This process can be called "decelerated swelling". At 816 h there was 28.1% of the total swelling and its average swelling rate was only 0.00014%/h, which is much smaller than the previous two stages. In addition, the change in the curve after T8 indicates that the intergranular swelling had basically ended, mainly due to the intergranular swelling caused by the adsorption of water film. As the water film thickened, the swelling potential energy of the particle surface decreased rapidly. At the same time, capillary condensation started to fill the intergranular pores, causing further intergranular swelling. Because the

capillary hydraulic pressure was obviously much weaker than the adsorption pressure of the intergranular and water film, the swelling rate initially declined but finally stabilized after entering the capillary condensed absorption stage.

**Table 4.** Intermittent moisture absorption-swelling law.

| Intermittent Phase | Duration (h) | Average Moisture Absorption Rate (%/h) | Time Point (h) |
|---|---|---|---|
| T1 | 1 | 0.0396 | 0 |
| T2 | 2 | 0.0285 | 9 |
| T3 | 6 | 0.0092 | 97 |
| T4 | 6 | 0.0053 | 198 |
| T5 | 7 | 0.0030 | 312.5 |
| T6 | 7 | 0.0014 | 364.3 |
| T7 | 13 | 0.0002 | 563.3 |
| T8 | 30 | 0.0003 | 701.6 |

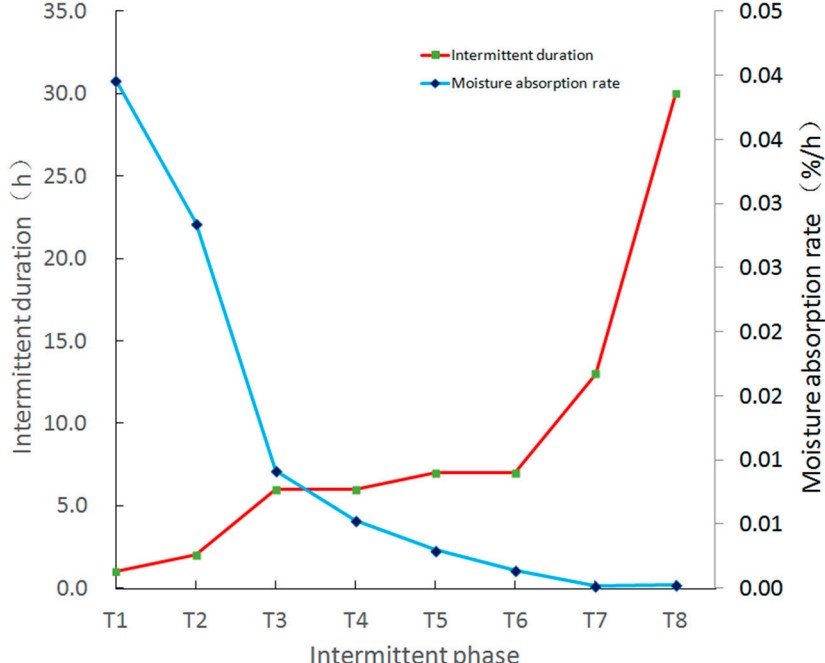

**Figure 8.** Intermittent time and average moisture absorption rate of each stage.

### 4.1.3. Coordination Analysis of Moisture Absorption and Swelling

Based on the above analysis, gaseous moisture absorption is the main cause of swelling. However, judging from the stage division of moisture absorption and expansion, the two are not completely synchronized. Table 5 lists the test data of each phase of the two. Figure 9 shows the evolution of the gaseous moisture absorption and swelling with time. It can be seen that moisture absorption and swelling were related to each other and there are certain differences. We will analyze this in the following sections.

In the first 84 h, both moisture absorption and swelling were rapid. The corresponding rapid moisture absorption basically coincided with the rapid swelling phase. On the one hand, water molecules were adsorbed between the clay crystal layers. The higher suction potential accelerated the moisture absorption ratio. On the other hand, the water molecular layer led to the swelling of particle volume, which in turn determined the larger swelling rate in the rapid moisture absorption stage controlled by the adsorption of the crystal layer.

**Table 5.** Test results of moisture absorption-swelling at each stage (99% *RH*).

| Moisture Absorption Stage | Duration (h) | Proportion of Time (%) | Moisture Absorption Ratio (%) | Proportion of Moisture Absorption (%) | Average Moisture Absorption Rate (%/h) |
|---|---|---|---|---|---|
| Rapid moisture absorption | 0~84 | 5.35 | 1.58 | 40 | 0.0190 |
| Decelerated moisture absorption | 84~320 | 15.03 | 3.15 | 40 | 0.0067 |
| Slow moisture absorption | 320~1570 | 79.62 | 3.94 | 20 | 0.0006 |
| Swelling stage | Duration (h) | Proportion of time (%) | Swelling strain (%) | Proportion of swelling (%) | Average swelling rate (%/h) |
| Rapid swelling | 0~84 | 5.43 | 0.064 | 15.7 | 0.00070 |
| Intermittent swelling | 84~730 | 41.78 | 0.290 | 56.2 | 0.00035 |
| Decelerated swelling | 730~1546 | 52.78 | 0.406 | 28.1 | 0.00014 |

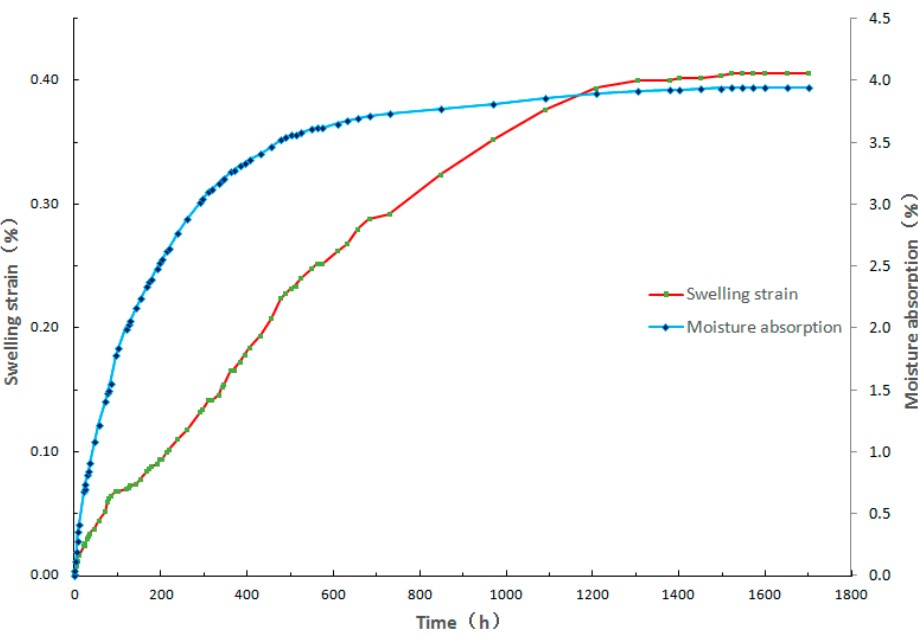

**Figure 9.** Comparison curve of gaseous moisture absorption and swelling with time.

Relative to the total moisture absorption ratio and swelling strain, the rapid moisture absorption stage reached 40% in the first 84 h, while the swelling stage only developed by 15.7%. This suggested that the latter lags significantly behind the former. The hysteresis effect was further amplified during the intermittent swelling stage, which can be quantified by the difference, $\Delta cp$, between the two percentages:

$$\Delta cp(t) = w(t)/w(\infty) - \delta(t)/\delta(\infty) \tag{5}$$

where $w(\infty)$ and $\delta(\infty)$ were the final equilibrium moisture absorption ratio and swelling strain, respectively.

The variation of $\Delta cp$ with time is presented in Figure 10. $\Delta cp$ reached its maximum at 320 h, corresponding to the transition point of the decelerated moisture absorption and slow moisture absorption stages. At this time, the moisture absorption reached 80%, while the swelling strain was only 36%. This hysteresis phenomenon implied that the migration of water to the interlayer, particle surface and intergranular pores is a gradual

process. There is a balance process between the swelling of the crystal layer, the thickening of water film and the change in the internal stress of particles and pores, which may be the internal mechanism that describes why the macroscopic deformation lags behind the moisture absorption.

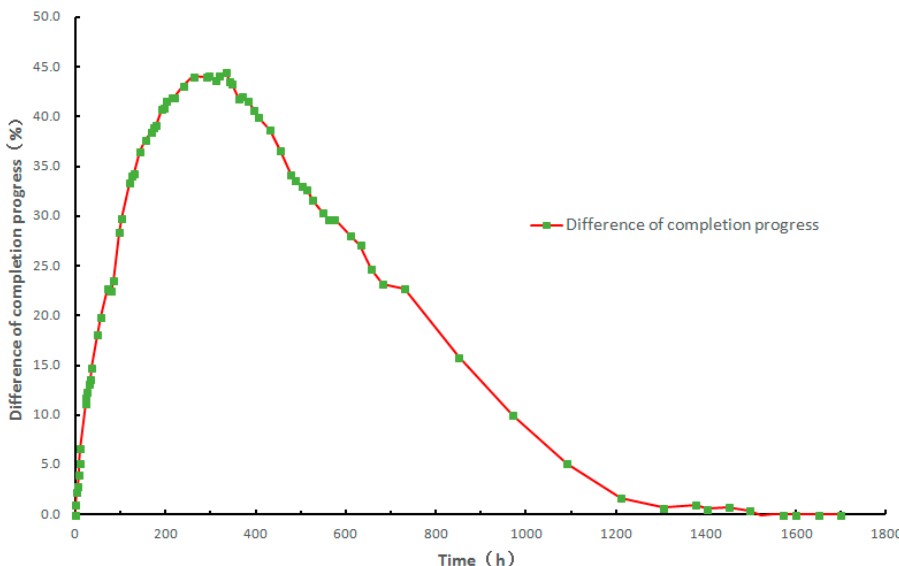

**Figure 10.** Variation of hysteresis parameter $\Delta cp$ with time.

After entering the slow moisture absorption stage, the deformation will continue for a period of time under the influence of the hysteresis effect. Then, the swelling rate also gradually decreased similarly to the moisture absorption rate. This is mainly for two reasons. One may be that due to the accumulation of moisture absorption, the water content of mudstone increased. The difference between it and the suction potential of gaseous water vapor gradually narrowed. Accordingly, the swelling potential energy was also weakened. The second may be that with the continuous volume change, the original pores and fracture spaces will be squeezed, thereby reducing the connectivity between pores to form partially closed pores. This will limit the further migration of moisture to the interior, so the moisture absorption rate will slow down over time. In summary, moisture absorption and swelling are two processes that influence and restrict each other but are not completely synchronized.

In Figure 10, under the interaction of moisture absorption and swelling, the two reached a balanced and stable state. This standard is judged according to the respective trends of the curves. Based on the results, the stability standard of red-bed mudstone can be specifically quantified as having a moisture absorption rate of <0.001 g/d and a swelling rate of <0.001 mm/d.

*4.2. Effect of RH on the Gaseous Moisture Absorption and Swelling Properties*

4.2.1. Gaseous Moisture Absorption Law at Different RHs

The experimental results of the gas moisture absorption of red-bed mudstone under different *RH* are presented in Figure 9. The characteristic curves had similar shapes, and both showed a decelerating growth trend. In this regard, the first-order exponential decay function can be used uniformly for fitting:

$$w(t) = w(\infty, RH) - w(\infty, RH)e^{-t/\beta} \qquad (6)$$

where $w(\infty, RH)$ is the final equilibrium moisture absorption ratio at a certain *RH* and $\beta$ is a fitting parameter related to the gaseous moisture absorption ratio (Table 6 and Figure 10).

**Table 6.** Moisture absorption ratio within 6 d before the test under different *RHs*.

| *RH*/% | Moisture Balance Time *t*/h | *w*(∞, *RH*)/% | *β* | *R*$^2$ |
|---|---|---|---|---|
| 99 | 1570 | 3.94 | 193.0 | 0.99 |
| 85 | 311 | 2.28 | 103.0 | 0.99 |
| 70 | 150 | 1.75 | 61.8 | 0.98 |
| 60 | 135 | 1.31 | 39.3 | 0.96 |

The increased *RH* in Figure 10 caused the equilibrium time (*t*) to become longer. If *t* = 150 h at 60% *RH* is viewed as the reference value, *t* at a *RH* of lower than 85% increased to about double the reference value. When the *RH* exceeded 85%, *t* enhanced significantly and expanded to about 10 times its value at a 99% *RH*.

The equilibrium moisture absorption ratio was positively correlated with the *RH*, specifically from 1.31% (*RH* = 60%) to 3.94% (*RH* = 99%), and approximately increased by three times. This is far greater than the growth rate of the equilibrium moisture absorption ratio, which indicates that within *RH* = 60%~99%, the non-linear characteristic of swelling rate changing with relative humidity was stronger than that of the moisture absorption rate. One possible explanation for this effect should be related to the water distribution and the sensitivity of clay minerals to water content. The above results also show that the lower the *RH*, the greater the suction and the smaller the difference between it and the suction potential of dry mudstone. Therefore, as the *RH* increased, the difference in suction potential further enlarged which could be the internal reason for the rise to the equilibrium moisture absorption ratio, equilibrium time length and moisture absorption rate.

The relationship between equilibrium moisture absorption ratio and *RH* can be described by the exponential equation (Figure 10):

$$w(\infty, RH) = 1.16 + 0.008 e^{RH/6.825} \tag{7}$$

The moisture absorption ratio is also directly proportional to *RH*. The change in *β* with *RH* can be fitted by the following formula:

$$\beta = -3.93 + 43.22 e^{(RH-60)/25.72} \tag{8}$$

Substituting Equations (7) and (8) into (6), the mathematical expression of the gaseous moisture absorption curve of the central Sichuan red-bed mudstone that is only related to *RH* can be obtained.

### 4.2.2. Variation of Swelling Strain at Different RHs

In Figure 11, the comparison results of swelling strain under four *RHs* are plotted. The swelling–time curves have similar characteristics. They all went through three stages of rapid swelling, intermittent swelling and decelerating swelling. The final stable swelling strain and stable duration under different *RHs* are shown in Table 7 and Figure 12.

**Table 7.** Swelling strain at different *RHs*.

| *RH*/% | *t′*/h | *δ*(∞, *RH*)/% |
|---|---|---|
| 99 | 1546 | 0.406 |
| 85 | 407 | 0.140 |
| 70 | 145 | 0.112 |
| 60 | 110 | 0.018 |

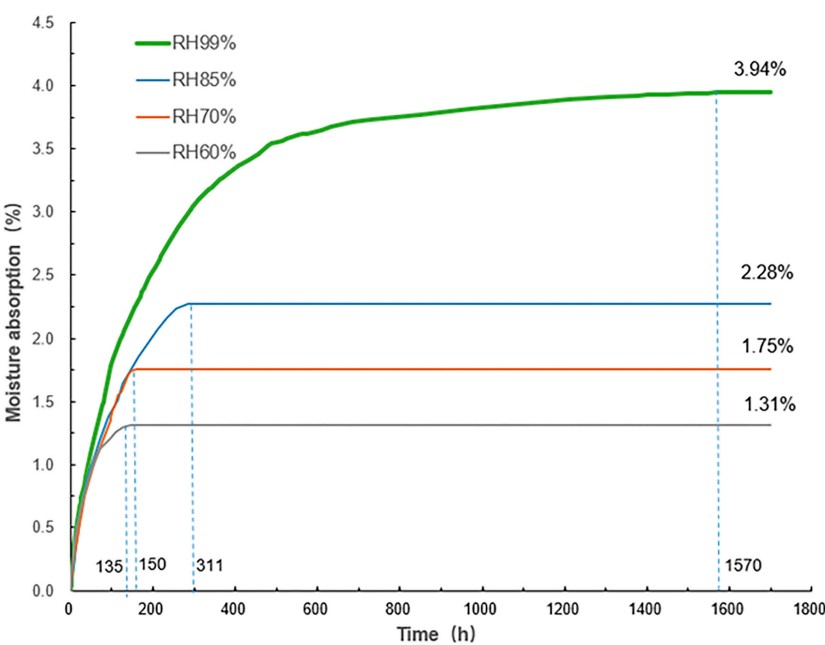

**Figure 11.** Comparison of moisture absorption characteristic curves of mudstone under different *RHs*.

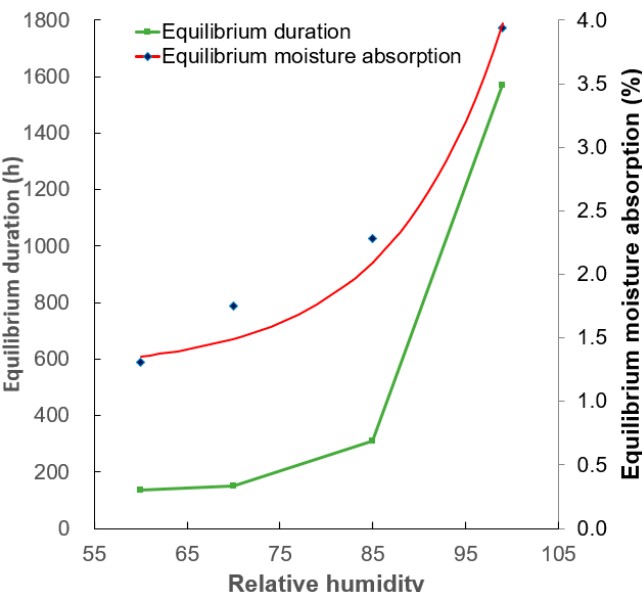

**Figure 12.** Changes in equilibrium moisture absorption ratio and equilibrium time with *RH*.

With the rise in *RH*, the swelling stable time (*t'*) was longer but slightly shorter than the equilibrium time (*t*) of moisture absorption, indicating that although the swelling lagged behind the moisture absorption, once the moisture absorption stops, the swelling will end. Likewise, *t'* changed a little when *RH* < 85%, but became significantly larger when *RH* > 85%.

The stable swelling strain was also proportional to *RH*, which increased by about 20 times from *RH* = 60 to 99%, which is far greater than the growth rate of the equilibrium moisture absorption ratio of about 3 times. It indicates that within the general range of humidity variation 60%~99%, the non-linear characteristic of expansion rate changing with relative humidity was stronger than that of the moisture absorption rate. The relationship between the stable swelling strain and *RH* can be reflected by the following equation (Figure 12):

$$\delta(\infty, RH) = 0.0286 + 0.000184 e^{RH/12.99} \qquad (9)$$

It is worth noting that the average swelling strain does not have a clear correlation with *RH* like the moisture absorption ratio does (Figure 13). The reason for this may be related to the discontinuous change in interlayer spacing. As listed in Table 8, the onset of deformation at each *RH* lagged behind that of the moisture absorption. For example, the duration of the first intermission, T1, was inversely proportional to the *RH*, indicating that the hysteresis of swelling was more pronounced. At the same time, the intermittent times decreased from 8 to 3 as the *RH* declined. The intermittence duration at the same *RH* also climbed non-linearly, which means that the decrease in *RH* reduced the difference in suction potential. When gaseous water molecules were adsorbed into the interlayer of clay crystals, the formed water molecule layer took longer to store energy and convert potential energy.

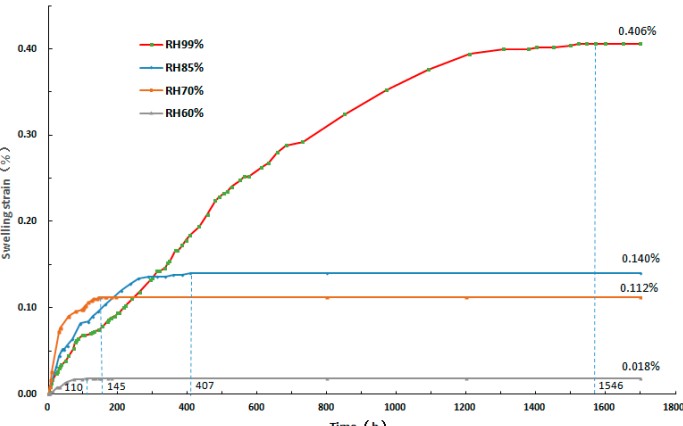

**Figure 13.** Swelling strain-time curve of red-bed mudstone under different *RHs*.

**Table 8.** Statistics of the intermittent times and duration of swelling.

| nth Intermittent Swelling | Intermittent Duration (h) | | | |
|---|---|---|---|---|
| | *RH* = 99% | *RH* = 85% | *RH* = 70% | *RH* = 60% |
| T1 | 1 | 1 | 2.2 | 2.5 |
| T2 | 2 | 3.9 | 4 | 6.1 |
| T3 | 6 | 14.5 | 15 | 23.7 |
| T4 | 6 | 47.5 | — | — |
| T5 | 7 | 23 | — | — |
| T6 | 7 | — | — | — |
| T7 | 13 | — | — | — |
| T8 | 30 | — | — | — |

Combined with Figures 12 and 14, an 85% *RH* was the turning point of moisture absorption ratio, swelling strain and equilibrium time. This is instructive for actual engineering because the measured data showed that the RH of the underground engineering environment mostly fluctuated around 85%. When the environmental *RH* < 85%, the process of the moisture absorption equilibrium of mudstone was relatively short and the corresponding swelling deformation was also small. When the *RH* > 85%, caused by environmental changes or artificial disturbances, the red-bed mudstone in central Sichuan will enter a long period of moisture absorption-swelling. Equilibration will take over two months and the deformation will increase significantly.

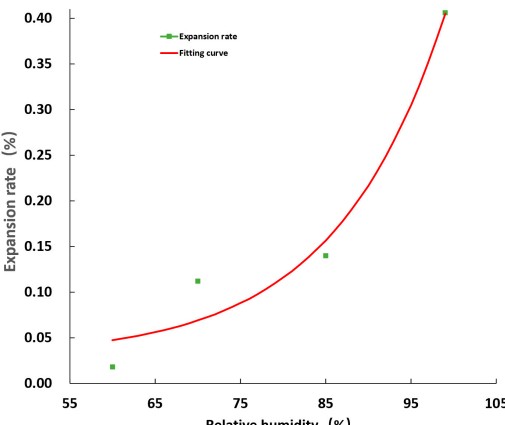

**Figure 14.** Evolution of swelling strain with *RH*.

### 4.3. Soil–Water Characteristic Curve of Red-Bed Mudstone under High Suction

The relationship of the suction under different RHs with the equilibrium water content (soil–water characteristic curve) is shown in Figure 15. The *RH* = 60%~99% set in the test basically represented the actual measurement range within the natural environment. The corresponding suction was 1.39~70.34 MPa. The soil–water characteristic curve in Figure 15 mainly corresponded to the gaseous moisture absorption equilibrium stage. If it is spliced with the soil–water characteristic curve under the condition of liquid moisture absorption (low suction level), a complete soil–water characteristic curve can be formed. This will provide key constitutive parameters for the numerical simulation of the moisture absorption-swelling of the mudstone in central Sichuan. Furthermore, the soil–water constitutive parameters of gaseous moisture absorption (high suction level) are particularly important for calculating the moisture field of mudstone under the influence of climate change.

$$u_a - u_w = 262.37e^{\left(-\frac{w}{1.07}\right)} - 5.99 \tag{10}$$

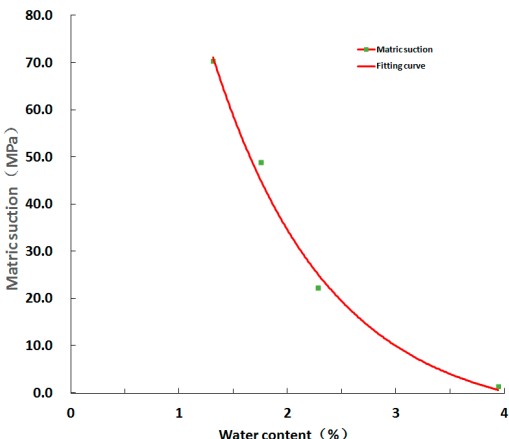

**Figure 15.** Soil–water characteristic curve under high suction.

In Figure 16, the relationship between the swelling strain $\delta$ of the red-bed mudstone and the increment of water content $\Delta w$ in the high suction environment is fitted. In the gaseous moisture absorption stage, the two were approximately linearly positively correlated. This can be expressed as follows:

$$\delta = 0.1436\Delta w - 0.1641 = 0.1436(w - w_0) - 0.1641 \tag{11}$$

where $w_0$ is the initial water content. If the drying state is taken as the starting point, then $w_0$ = 0. This formula establishes a bridge between the volumetric strain field and the

humidity field, which can provide a basis for the deformation calculation of the red-bed mudstone in central Sichuan under different environmental humidities.

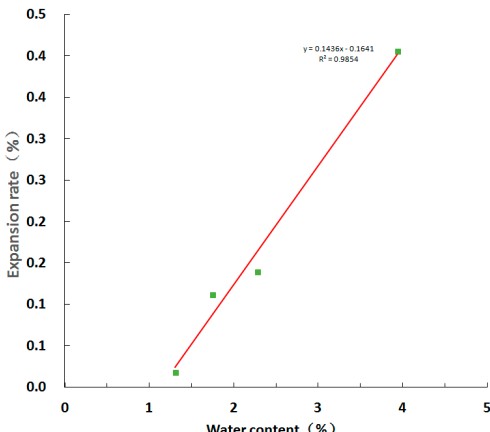

**Figure 16.** Fitting curve of swelling strain and increment of water content at high suction level.

## 5. Conclusions

The gaseous moisture absorption-swelling test under different *RHs* was carried out for red-bed mudstone in central Sichuan, China. By analyzing the time-varying characteristics of gaseous moisture absorption and swelling, as well as the coordination of the two, the evolution of it over time and its internal mechanism were clarified. The key conclusions are as follows:

(1) The gaseous moisture absorption of red-bed mudstone was a long-term and slow process. Under the condition of a 99% *RH*, it took more than 2 months for the sample to be stable. According to the change in moisture absorption ratio, it can be roughly divided into three stages: the rapid moisture absorption stage dominated by the adsorption of clay crystal layers, the decelerated moisture absorption stage dominated by the water film on the particle surface and the slow moisture absorption stage dominated by capillary condensation in the intergranular pores.

(2) The gaseous moisture absorption of mudstone was accompanied by an obvious deformation. Intermittent swelling was an important feature which may be related to the discontinuous change in interplanar spacing. The intermittent duration was inversely proportional to the *RH*, while the number of intervals was directly proportional to it.

(3) Gaseous moisture absorption and swelling are two processes that influence and restrict each other but are not synchronized. The swelling lagged significantly behind moisture absorption and this hysteresis effect reached a maximum at the transition from decelerated to slow moisture absorption. At this time, the corresponding moisture absorption ratio and swelling rate were 80% and 36%, respectively.

(4) The equilibrium time, moisture absorption ratio and swelling strain all enhanced nonlinearly with the rise in *RH*. The characteristic point of the hygroscopic expansion change corresponds to *RH* = 85%. By fitting the experimental data, the soil–water characteristic curve of mudstone under a high suction level was obtained.

The findings are crucial for comprehending the intricate interactions between red-bed mudstone and moisture. The research outcomes have wide-ranging applications in diverse engineering scenarios, including tunnel construction, underground excavation and slope stabilization. Particularly, they offer valuable insights in identifying and addressing potential engineering challenges arising from moisture-induced deformations in mudstone.

**Author Contributions:** Methodology, F.Y. and Z.F.; formal analysis, F.Y. and J.L.; writing—original draft, F.Y., K.T. and Z.F.; funding acquisition, F.Y. and Z.D.; supervision, K.T. and Z.D.; data curation, J.L. and K.H. All authors have read and agreed to the published version of the manuscript.

**Funding:** This research work was funded by the National Natural Science Foundation of China (No. 42172308, funder: Zhangjun Dai) and the Youth Innovation Promotion Association CAS (No. 2022331, funder: Zhangjun Dai).

**Institutional Review Board Statement:** Not applicable.

**Informed Consent Statement:** Not applicable.

**Data Availability Statement:** The datasets involved in the current study are available from the corresponding author, K.T. (tongkaiwen19@mails.ucas.edu.cn), upon reasonable request.

**Conflicts of Interest:** The authors declare no conflict of interest.

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
