# Peer review of "Experimental Study on Gaseous Moisture Absorption and Swelling of Red-Bed Mudstone in Central Sichuan, China under Different Relative Humidity Environments"

_sustainability, doi:10.3390/su151512063_

Round 1

Reviewer 1 Report

The manuscript investigates the gradual development process of moisture absorption and expansion in the gas phase of the Sichuan red-bed mudstone under different relative humidity conditions. It elucidates the dominant factors influencing the moisture absorption rate changes at each stage and the intrinsic mechanism of intermittent expansion. Furthermore, it clarifies the nonlinear influence of relative humidity on equilibrium duration, moisture absorption rate, and expansion rate. The study also establishes the soil-water characteristic curve and the quantitative relationship between gas-phase moisture absorption expansion rate and incremental moisture content for the Sichuan red-bed mudstone at a high suction level. The manuscript presents innovative insights obtained from the experiments and combines them with theoretical analysis to provide a direct basis for analyzing long-term micro-expansion of the Sichuan red-bed mudstone under environmental influences. The main comments are as follows:

1)       In the abstract, specify in which stage intermittent expansion primarily occurs and the controlling factor.

2)       Clarify the natural occurrence conditions of the mudstone samples with undeveloped primary fissures used in the experiments.

3)       Supplement information on the methodology employed to achieve unidirectional expansion in the tests, including the size and dimensions of the side wall moisture absorption holes.

4)       Provide an explanation for adopting the limited displacement method to measure the expansion force and highlight the differences from the traditional backpressure method.

5)       Since the principle and maturity of the electrolyte saturation solution humidity control method are well-established, consider deleting the relevant introduction to its principles.

6)       Include the Young-Laplace equation for capillary condensation phase change.

7)       Explain the significance and reason behind the gas-phase moisture equilibrium content (3.94%) being higher than the natural moisture content (2.77%).

8)       Supplement the information on the magnitude of the nonlinear increase in moisture absorption rate with relative humidity and provide a possible explanation for why relative humidity has a more pronounced effect on the expansion rate than the moisture absorption rate.

The manuscript investigates the gradual development process of moisture absorption and expansion in the gas phase of the Sichuan red-bed mudstone under different relative humidity conditions. It elucidates the dominant factors influencing the moisture absorption rate changes at each stage and the intrinsic mechanism of intermittent expansion. Furthermore, it clarifies the nonlinear influence of relative humidity on equilibrium duration, moisture absorption rate, and expansion rate. The study also establishes the soil-water characteristic curve and the quantitative relationship between gas-phase moisture absorption expansion rate and incremental moisture content for the Sichuan red-bed mudstone at a high suction level. The manuscript presents innovative insights obtained from the experiments and combines them with theoretical analysis to provide a direct basis for analyzing long-term micro-expansion of the Sichuan red-bed mudstone under environmental influences. The main comments are as follows:

1)       In the abstract, specify in which stage intermittent expansion primarily occurs and the controlling factor.

2)       Clarify the natural occurrence conditions of the mudstone samples with undeveloped primary fissures used in the experiments.

3)       Supplement information on the methodology employed to achieve unidirectional expansion in the tests, including the size and dimensions of the side wall moisture absorption holes.

4)       Provide an explanation for adopting the limited displacement method to measure the expansion force and highlight the differences from the traditional backpressure method.

5)       Since the principle and maturity of the electrolyte saturation solution humidity control method are well-established, consider deleting the relevant introduction to its principles.

6)       Include the Young-Laplace equation for capillary condensation phase change.

7)       Explain the significance and reason behind the gas-phase moisture equilibrium content (3.94%) being higher than the natural moisture content (2.77%).

8)       Supplement the information on the magnitude of the nonlinear increase in moisture absorption rate with relative humidity and provide a possible explanation for why relative humidity has a more pronounced effect on the expansion rate than the moisture absorption rate.

Author Response

The revised manuscript and the response to the reviewer’s comments have been uploaded, please see the attachment.

Reviewer 2 Report

Red bed mudstone is prone to disintegration under the influence of water, and has geological characteristics such as severe strength attenuation, strong hydrophilicity and weak permeability, which has caused many engineering problems in the world, which makes the study of the influence of water on mudstone become a hot topic in engineering construction. In this paper, a gas hygroscopic expansion test device is designed for the long-term microexpansibility of red-bed mudstone, and the gas hygroscopic expansion test under four kinds of ambient humidity is carried out. The influence of vapor pressure gradient on equilibrium time, moisture absorption rate and expansion rate is discussed, and the related mechanism of gas moisture absorption and expansion is explained. The thesis topic has a good idea, and the research conclusion has a certain theoretical value and engineering practicability. The logic of the paper is good, but there are a few deficiencies still need to be modified and perfected. The specific problems are as follows:

 (1) The abstract should include the purpose, methods, results and conclusions of the research, and the abstract says that the experimental process is long-term and slow, may I ask what is the basis for the judgment of long-term and slow, suggest the author to further improve and modify the abstract of this paper.

(2) For the study of red-bed mudstone, the reference sources in the research status in the introduction part of the article should be more extensive.

(3) Some pictures in the article are not clear, and the annotation text in some pictures is small, the color of the text does not match, etc. It is suggested to change some pictures. For example, Figure 4-Figure 8, Figure 15-Figure 16, etc.

(4) I am very interested in the hygroscopic expansion test device in the third part of this paper. May I ask what errors affect the quality of rock samples measured by this device? How do you consider air pressure, buoyancy, gravity, temperature and other environmental factors when designing it?

(5) "Figure 13. Swelling strain-time curve of red-bed mudstone under different RH." and "Figure 14. Evolution of the fourth part of the article swelling strain with RH. "Respectively from two angles to explain the change of moisture absorption swelling, should be expressed with the corresponding picture.

(6) The conclusion needs to be further improved. For example, "There was almost no change when RH<85%, but they increased rapidly when RH>85%" in conclusion (4). There is ambiguity that should be defined as hygroscopic expansion change.

 In conclusion, the experimental method and design of this paper are reasonable, the research content is substantial, and the research conclusion is reliable. It is recommended to be accepted after modification!

Author Response

(The authors gave the same response as above.)

Reviewer 3 Report

In spite of its relatively limited innovation, the paper entitled "An Experimental Investigation of Gaseous Moisture Absorption and Swelling of Red-bed Mudstone in Central Sichuan, China under Various Relative Humidity Environments" can be considered for publication if certain modifications are made. Changes suggested include:

Identifying the Uniqueness of the Research: It is crucial that your research be distinguished from those previously conducted. Specify the distinguishing aspects, such as methodology, experimental setup, or data analysis techniques employed, which set your research apart.

Explicit Problem Statement and Significance: The problem statement should be clearly articulated and the significance of the research should be highlighted. You should describe the specific issue or knowledge gap that your study addresses and how its resolution contributes to the existing body of knowledge in the field. 

SEM Analysis in Figure 2: Enhance the clarity of Figure 2, which shows the SEM (Scanning Electron Microscope) analysis. In order to facilitate better understanding, highlight any visible cracks in the figures and label them appropriately. Provide a reason why  SEM analysis for samples with different water contents is not shown. 

Rectify Table 2: Correct the confusion regarding the minimum and maximum values presented in Table 2. The minimum values should always exceed the maximum values, as this inconsistency undermines the reliability and credibility of the data. Make sure the table accurately represents the results of your experiment.

Moderate changes are required. 

Author Response

(The authors gave the same response as above.)

Round 2

Reviewer 3 Report

I am happy to accept the revised version.